# Numerical and Experimental Investigation of the Effect of Current Density on the Anomalous Codeposition of Ternary Fe-Co-Ni Alloy Coatings

**DOI:** 10.3390/ma15176141

**Published:** 2022-09-04

**Authors:** Shuai Zhang, Jing Yu, Zhengda Liu, Yanjun Yin, Chenfeng Qiao

**Affiliations:** 1Marine Engineering College, Dalian Maritime University, Dalian 116026, China; 2China North Engine Research Institute Tianjin, Tianjin 300400, China; 3Department of Materials Science and Engineering, Dalian Maritime University, Dalian 116026, China

**Keywords:** numerical simulation, anomalous codeposition, Fe-Co-Ni alloy coating, current density

## Abstract

Gradient-structured ternary Fe-Co-Ni alloy coatings electrodeposited on steel substrates at various current densities from chloride baths were numerically and experimentally investigated. The electrodeposition process, considering hydrogen evolution and hydrolysis reaction, was modelled using the finite element method (FEM) and was based on the tertiary current distribution. The experimentally tested coating thickness and elemental contents were used to verify the simulation model. Although there was a deviation between the simulation and experiments, the numerical model was still able to predict the variation trend of the coating thickness and elemental contents. The influence of the current density on the coating characterization was experimentally studied. Due to hydrogen evolution, the coating surface exhibited microcracks. The crack density on the coating surface appeared smaller with increasing applied current density. The XRD patterns showed that the deposited coatings consisted of solid-solution phases α-Fe and γ (Fe, Ni) and the metallic compound Co_3_Fe_7_; the current density in the present studied range had a small influence on the phase composition. The grain sizes on the coating surface varied from 15 nm to 20 nm. The microhardness of the deposited coatings ranged from 625 HV to 655 HV. Meanwhile, the average microhardness increased slightly as the current density increased from 5 A/dm^2^ to 10 A/dm^2^ and then decreased as the current density further increased. Finally, the degree of anomaly along with the metal ion and hydrogen atom concentrations in the vicinity of the cathodic surface were calculated to investigate the anomalous codeposition behaviour.

## 1. Introduction

Electrodeposition is usually a more cost-effective surface coating technology than other coating fabrication methods, such as physical and chemical vapour deposition. It is particularly effective for fabricating alloy layers on complex geometries [1,2,3]. Electrochemically deposited alloy coatings can prolong the service life of industrial applications and even extend the application scope of substrate metals, thereby reducing production cost [4,5]. Ternary Fe-Co-Ni alloy coatings possess a wider spectrum of properties that have attracted extensive attention in industrial areas and in research, such as good soft magnetic properties [6], high hardness, anticorrosion [7], and wear resistance [2,4,8]. Depending on the composition of Fe-Co-Ni alloys, the as-deposited coating can exhibit various properties. It is known that the use of galvanic Fe-Co-Ni alloy coatings that are rich in Ni and Co leads to the enhancement of electric and magnetic performance [9], while electrodeposited coatings rich in Fe exhibit high wear resistance [10]. To date, iron group ternary alloy thin films that are rich in Co and Ni have been extensively studied and used in the electromagnetic industry. However, few works have focused on thick Fe-Co-Ni alloy coatings rich in Fe, which have great potential for use in protective coatings to provide corrosion and wear resistance.

Previously, our research team successfully electrodeposited a binary Fe-based alloy coating with a gradient structure that transitioned from coarse columnar grain to nanocrystalline. This coating is deposited using alternating current and direct current in sequence [11]. The adhesion strength of the as-deposited coating was as high as approximately 400 MPa, as tested using a modified Ollar method [12]. Moreover, due to the low pH value in the electrolyte and the high applied current density, a reticular coating surface was generated, which was able to entrap the wear debris, reserve the lubrication oil, and relieve the inner stress.

Fe, Co, and Ni (iron, cobalt, and nickel) are iron group metals, and their codeposition presents an anomalous phenomenon. The less noble metal (Fe or Co) is deposited preferentially to the more noble metal (Ni) [13,14]. To date, the mechanism for iron-group binary alloy anomalous codeposition has been extensively investigated [15], including modelling the reaction mechanism [16], modulating the electrodeposition parameters [17], and using various electrochemical measurements [18]. The anomalous codeposition of the Fe-Co-Ni ternary alloy is an irreversible two-step reaction mechanism. However, there are two different viewpoints in terms of intermediates. Zhuang et al. suggested that during the electrodeposition process, single-metal absorbed species of Fe(I)_ad_, Co(I)_ad_, and Ni(I)_ad_ along with mixed metal intermediate species of FeNi(III)_ad_, FeCo(III)_ad_, and NiCo(III)_ad_ covered the cathode surface [19]. They postulated that the more noble metal ion Ni^2+^ plays the role of a catalyst in order to enhance the codeposition of the less noble metal ions Fe^2+^ and Co^2+^. In contrast, Dahms [20] proposed a general mechanism for anomalous codeposition, in which the hydrolysis effect of the solvent (H_2_O) was taken into account. The authors pointed out that metal hydroxides are the primary intermediate products that can suppress the deposition of more noble metals but facilitate the discharge of less noble metals. Li et al. [21] analysed the polarization curves of anomalous codeposition and suggested that iron monohydroxyl shielded the cathode surface, which inhibited the discharge of cobalt and nickel hydroxide while activating the discharge process of iron monohydroxyl. Note that the intermediate species depend on the electrolyte composition and operation conditions. The above investigations suggest that the mechanism of Fe-Co-Ni anomalous codeposition is intricate and that the existing theoretical models cannot explain the experimental phenomenon and anomalous behaviour very clearly.

Multiphysical field coupling simulations have been extensively used in electrochemical deposition studies [22]. Because of the savings provided in time and cost, this simulation method can optimize the deposition conditions and visualize the experimental phenomena, which are rarely observed. The combining of an experiment with simulations can help us to further understand the behaviour and the mechanism behind the phenomenon. Fu et al. [23] established a numerical model to simulate the jet electrodeposition process at different scanning speeds. Using commercial COMSOL software, Zhang et al. [24] studied the behaviour of Ce^3+^ ions in the electrodeposition of LiCl-KCl molten salt, the cathodic reaction process, and the factors influencing the deposited layer thickness. Khazi et al. [25] simulated the influence of magnetic stirring speed on the velocity within the electrolytic cell and optimized the layout of the electrodes. Manzano et al. [26] explored the potential application of electrodeposition in complex geometries and various dimensions and built a three-dimensional model using COMSOL. On the basis of a two-dimensional numerical model, Maraschky et al. [27] compared the degree of coating uniformity electrodeposited by pulsed current and direct current. Yang et al. [28] optimized the layouts of anodes and cathodes to achieve electrodeposition thickness uniformity through numerical simulation and experiments. Yue et al. [29] analysed the influence factors for coating thickness based on the surrogate model and experimental results. Belov et al. [30] reported that, for the electrodeposition process with either a low flow velocity of electrolyte or without agitation, the coating thickness can be predicted precisely by the tertiary current distribution considering mass transport. Although many scholars have developed numerical models for electrodeposition, a simplified one-step electrochemical reaction was used in the above simulations. However, for the electrodepostion of ternary Fe-Co-Ni alloys, the reaction process is performed in an irreversible two-step manner. In addition, the side reactions of hydrogen evolution and water dissociation significantly influence the properties of deposits. To the best of our knowledge, there is currently no specific simulation study for the Fe-Co-Ni ternary alloy deposition.

Among the various electrodeposition parameters, current density is the most influential factor affecting the surface morphology, elemental composition, and properties of the electrodeposited layer [31,32]. In this study, the electrodeposition of ternary Fe-Co-Ni alloy coatings was investigated using numerical simulations and experiments. The coating thickness and elemental composition were examined to verify the accuracy of the numerical model. Additionally, the influence of the applied current density on the coating morphology, phase composition, grain size, and microhardness was examined. Finally, we investigated the anomalous codeposition behaviour on the basis of the degree of anomalous codeposition and ion concentration near the cathodic surface.

## 2. Experimental Procedure for Electrodeposition

### 2.1. Materials and Electrodeposition

42CrMo alloy steel was selected as the cathode material. This steel is commonly used in crankshafts, gears, and connecting rods. The industrial pure iron strip Q235 was used as the soluble anode. In industry practice, a number of anodes are placed in an electroplating bath to improve the thickness uniformity of components [33]. In the present study, four anodes were laid out at each corner of the electrochemical cell. Four round specimens of 42CrMo were welded together in a row and immersed in the electrolyte. The dimensions of each specimen were Φ40 × 5 mm. The strip of Q235 and the welded 42CrMo specimens were hung on the electrodes through homemade holding fixtures. The experimental setup is shown in Figure 1.

The metal salt solution contained 100 g/L FeCl_2_·4H_2_O, 40 g/L CoCl_2_·7H_2_O, and 100 g/L NiCl_2_·6H_2_O. The pH was adjusted to approximately 0.4 to 0.6 with the hydrochloric acid. The electrodeposition processes were conducted at a temperature from 40 to 45 °C. A self-designed power supply, which can supply alternating current with a frequency of 25 Hz and direct current, was applied to provide an electric current. The whole electrodeposition process included activation with alternating current and coating electrodeposition with direct current, as shown in Figure 2. The purpose of the activation was to clean the specimen (cathode) surface and to facilitate the formation of metallic bonding between the substrate and coating. The direct current deposition process consisted of three stages in sequence: low current deposition, transitory deposition, and high current deposition. In this study, we adjusted only the high current stage in order to investigate the influence of the current density. It is noteworthy that the Fe-Co-Ni alloy coatings were deposited at an average current density from 5 to 25 A/dm^2^.

### 2.2. Electrodeposited Coating Analysis

According to the international standard [34] and industrial practice, the electric current detours along the component, consequently resulting in a large thickness close to the edge, namely, the edge effect. Therefore, the area that was 5 mm from the edge was avoided during the thickness measurement. The thickness was measured using an optical microscope (OM) at the cross-section of the deposited coating and a thickness gauge. The morphology and elemental content were examined using scanning electron microscopy (SEM) coupled with energy dispersive spectrometry (EDS). The phase composition was determined using X-ray diffraction (XRD), and the average grain size of the deposited coating was calculated using the Debye–Scherrer equation. The microhardness of the coating surface was measured using a Vickers hardness tester. The test conditions were as follows: the load was 200 g, and the loading time was 20 s. Five replicates were conducted for each specimen.

To verify the accuracy of the simulation model, the experimentally measured coating thickness and the elemental component were compared against the numerically calculated results. The thickness measurement position is shown in Figure 1. Furthermore, for EDS tests, we examined four specimens for every experiment and three points in each specimen and presented the average value of the elemental content.

## 3. Modelling of the Electrodeposition

### 3.1. Basic Assumptions

The commercial finite element analysis software COMSOL 6.0 Multiphysics was employed to calculate the thickness and elemental content of the deposited coatings and to simulate the metal ion concentration near the cathode surface under different applied current densities. The following assumptions were made:Biphasic electrolyte flow was not taken into consideration in the simulation model [35].The electrolyte was assumed to be the electroneutral [36]. The expression used is as follows:
(1)$$\sum_{m}z_{m}c_{m}=0$$
where the subscript *m* refers to electroactive constituents included in the simulation model.
3.The walls of the electrochemical cell were assumed to be insulated, and the calculation domain was an isothermal system.4.There was no extra stirring in the electrolyte, and the convection generated by hydrogen escaping along the electrodes was neglected [31].5.Based on the existing studies concerning the anomalous deposition of binary and ternary alloys [16,37], the following reactions were assumed to occur in this simulation:
(2)$$\mathrm{Fe}\left(\mathrm{II}\right)+M\left(\mathrm{II}\right)+e^{-}\to \mathrm{FeM}{\left(\mathrm{III}\right)}_{\mathrm{ad}}$$
(3)$$\mathrm{FeM}{\left(\mathrm{III}\right)}_{\mathrm{ad}}+e^{-}\to \mathrm{Fe}+M\left(\mathrm{II}\right)$$
(4)$$\mathrm{Co}\left(\mathrm{II}\right)+\mathrm{Ni}\left(\mathrm{II}\right)+e^{-}\to \mathrm{CoNi}{\left(\mathrm{III}\right)}_{\mathrm{ad}}$$
(5)$$\mathrm{CoNi}{\left(\mathrm{III}\right)}_{\mathrm{ad}}+e^{-}\to \mathrm{Co}+\mathrm{Ni}\left(\mathrm{II}\right)$$
where M represents Ni or Co.


### 3.2. Governing Equations

Within the proposed simulation domain, in addition to solving for the potential and current density in the electrodes and electrolyte, the metal ion concentration gradient and Faraday reaction were considered in the model. Meanwhile, the diffusion and migration of the metal salt in the solution were taken into consideration in the simulation model. Therefore, the tertiary current distribution was able to achieve relatively higher accurate simulation results [30]. The related mathematical model for the Fe-Co-Ni ternary alloy electrodeposition and the related parameter explanations along with the initial conditions are given in Table 1 and Table 2, respectively.

To investigate the concentration gradient of the metal ions near the electrode surface, a dilute solution approach [38] was employed. Because of the neglect of convection along the electrodes, the ion species flux in electrolyte *N_m_* can be described by the simplified Nernst–Planck equation:(6)$$N_{m}=-D_{m}\nabla c_{m}-z_{m}u_{m}Fc_{m}\nabla \phi_{l}$$
where $u_{m}$ is the ionic mobility derived from $u_{m}=D_{m}/kT$, k is the Boltzmann constant, $D_{m}$ is the diffusion coefficient, $c_{m}$ is the ion concentration, *z_m_* is the ion species valency, and $\phi_{l}$ is the potential in the electrolyte.

The material balance equation for ionic species at a certain position within the diffusion layer is described by:(7)$$\frac{dc_{m}}{dt}+\nabla \cdot N_{m}=R_{m}$$
where $R_{m}$ is the production rate of species m due to homogeneous chemical reactions. By substituting Equation (6) into (7), the ionic concentration within the diffusion layer can be obtained.

According to Faraday’s law [28], the thickness of the coating is calculated by [36]
(8)$$s_{tot}=\sum_{m}\frac{-j_{loc,m}}{zF}\frac{M_{m}}{\rho_{m}}t$$
where $M_{m}$ is the molar mass and $\rho_{m}$ is the density.

The atomic fraction of metal species (Fe, Co, Ni) in the deposited coating is calculated by


(9)
$$M(\mathrm{at}\%)=\frac{S_{tot,m}\rho_{m}/M_{m}}{\sum_{m}S_{tot,m}\rho_{m}/M_{m}}$$


### 3.3. Boundary Conditions and Mesh

The detailed boundary conditions on the research objects are listed in Table 3. A grid correlation analysis was conducted by adjusting the mesh size to guarantee the calculation accuracy and to meet the convergence requirements. Meanwhile, to consider the calculation time, the calculation domain was meshed with 72,353 triangular elements. In addition, the mesh in the electrode–electrolyte interfaces was refined using an extremely fine tetrahedral grid, as shown in Figure 3.

## 4. Results and Discussion

### 4.1. Simulating Results

The thickness profiles of the deposited coatings at different current densities are shown in Figure 4a. The thickness increased dramatically and caused remarkable inhomogeneity to evolve as the applied current density increased. Owing to the electric current detour around the cathode [42], the current density at the edge was higher than that in the middle. According to Faraday’s law, the uniformity of the deposition thickness is determined by the current density, and thus, a relatively high current density leads to greater changes in the coating thickness [28,43]. Figure 4b compares the differences between the simulation and the experiment. The simulation results were higher than the OM-measured results and lower than the gauge-measured results. This trend is similar that in [36], where the authors explained the deviation between the micrograph and gauge. The reasons for the differences between the simulation and experiment can be explained as follows: because a high applied current density leads to severe hydrogen evolution on the cathode, stirring caused by hydrogen gas inhibits the electrodeposition process [44] and thus reduces the coating thickness. In the present simulation model, the neglect of the convection along the electrodes and hydrogen adsorption on the cathode surface could potentially result in a difference between the calculated and measured thicknesses [45]. On the other hand, the electrochemical double layer could also have caused the deviation between the simulated and tested coating thicknesses. Some studies have [46,47] proven that a lower applied current density can reduce ion diffusion, while a higher applied current density can promote ion diffusion. However, at present the simulation model does not address the influence of the double layer on the coating thickness [36].

The simulated elemental compositions in the deposits at different applied current densities are shown in Figure 5a–e. Because of the exclusion of convention in the electrolyte and the biphasic effect in the numerical calculation, the elemental content distribution in the deposited coatings was relatively homogeneous, and the elemental contents varied within a reasonable range. A comparison between the simulation and the measured elemental contents is plotted in Figure 5f. The simulated results were very close to the elemental content tested using EDS. Moreover, with an increase in the applied current density, the Fe content in the deposited coatings increased, while the Co content decreased, and the Ni content increased slightly. The cathodic polarization increased with an increase in applied current density, and the cathodic potential shifted toward more negative potentials [48], which were close to the deposition potential of Fe. Consequently, the Fe content increased in the deposits. Additionally, the dominant effect of the cathodic process transformed the electrochemical reaction into diffusion control. Therefore, concentration polarization occurred in the vicinity of the cathode surface, and thus, the Co content decreased [49]. In general, the elemental content changed little at relatively high current densities [50].

On the other hand, the simulated Co content was slightly higher than the tested values, which can be attributed to the insulation and uniform temperature within the calculation domain [26]. In contrast, the simulated Fe content was slightly lower than the measured values. The neglect of convention and the diffusion constant [51] most likely resulted in the insufficient mass transmission of Fe^2+^ generated from soluble anodes. Although there are some assumptions and simplified boundary conditions within the model, the simulation model was still able to predict the variation trends in the coating thickness and the elemental contents in deposits as well as to facilitate the electrodeposition parameter design.

### 4.2. Experimental Results

Figure 6 shows the influence of the current density on the morphology of the cross-section and the coating surface of the ternary Fe-Co-Ni alloy coatings. The cross-sections of the Fe-Co-Ni ternary alloy electrodeposited coatings at various applied current densities are shown in Figure 6a-1–e-1. There was clearly no obvious interface between the substrate and the electrochemically deposited coating, which indicated that metallic bonding occurred between the metal matrix and the coating. After the AC activation stage, the mildly dissolved cathode surface promoted metal ions (Fe^2+^, Ni^2+^ and Co^2+^) accepting electrons and reduction [11]. As a result, metallic bonding occurred at the coating-substrate interface, which enhanced the adhesion of the ternary alloy coating to the metal matrix.

The cross-sectional micrographs captured using OM are shown in Figure 6a-2–e-2. Because of changes in the current mode, the grain size and microstructure of the deposited coating changed from the substrate-coating interface to the coating surface. It can also be observed that many discontinuous microcracks with black colouring were distributed within the coatings and grew perpendicular to the substrate surface. The cracked coating surfaces are shown in Figure 6a-3–e-3. The reasons for the microcrack generation can be explained as follows: the bonding strength across the grain growth direction was weaker than the bonding strength found along the growth direction, which was perpendicular to the substrate surface [52]. Owing to hydrogen penetration, the internal stress increased as the thickness of the coating increased. When the internal stress became high enough, the grain or grain boundary was torn. Consequently, numerous small cracks were generated. Coating cracking can decrease the internal stress [53], and consequently lead to an increase in the adhesion strength [54]. During electrodeposition, intermediates or impurities were entrapped in the cracks, resulting in the cracks appearing in a black colour. Our previous study found that the impurities were mainly composed of iron oxides or iron hydroxides [55]. Under a relatively low current density, the coating is thin. Some long cracks may extend through the whole deposited coating; therefore, the cracked surface presented with a higher crack density. As the applied current density increased, the crack length became shorter, and the crack width became narrower. Hence, the crack density on the coating surface seemed lower.

The XRD patterns for the Fe-Co-Ni deposited coatings are plotted in Figure 7. Fe-Co-Ni alloy coatings electrochemically fabricated at all applied current densities consist of the solid-solution phases α-Fe and γ(Fe, Ni) and the metallic compound Co_3_Fe_7_, which are the same as FeNi and FeCo binary alloy electrodeposits [56], respectively. It has been proven that bcc (body centred cubic) α-Fe is a stable phase for Fe content over 80% at temperatures below 500 °C [50]. The coexistence of two solid-solution phases in electrodeposition is common due to the nonequilibrium feature [57]. Furthermore, because of the high Fe content and the similar atomic radii for iron (1.72 Å), cobalt (1.67 Å), and nickel (1.62 Å), α-Fe is supposed to be the substitutional solid solution that can incorporate cobalt and nickel into iron. In addition, the texture component (211) is present in the deposited coating, which is a preferential orientation generated by electric crystallization [58]. Note also that the current density in the present studied range had a small influence on the relative peak intensities, which indicates the approximately constant phase composition of the deposited coatings.

The average crystallite sizes were calculated on the basis of the Scherrer equation using the full width at half maximum (FWHM) of the most intense diffraction peak [59], as shown in Figure 8. The deposited coating surface consisted of nanocrystallines with grain sizes of 15–20 nm; these results are very close to those determined using TEM analysis [50]. Some authors [60] reported that the grain size decreases with increasing applied current density for the iron group metal electrodeposition. However, in the present study, there was no clear relationship shown between the grain size and current density. Figure 8 also shows that the microhardness of the deposited coatings ranged from 625 HV to 655 HV. Meanwhile, the average microhardness increased slightly as the current density increased from 5 A/dm^2^ to 10 A/dm^2^, and then decreased as the current density further increased. Some studies [61] have observed that the microhardness of very fine grains does not align well with the Hall–Petch relationship. For the electrodeposited coating, the applied current density has a dominant influence on internal stress and hydrogen evolution, which in turn impacts the microhardness [17,62]. A higher internal stress [63] and the intensity of the hydrogen reaction were found to reduce microhardness. Moreover, the microhardness of the electrodeposited coating was positively correlated with the cobalt content [64]. From the previous EDS results, the Co content decreased with increasing current density, thus resulting in a decrease in microhardness. In addition, microstructures such as dislocations and pile-ups could contribute to the inverse Hall–Petch relationship [61].

### 4.3. Anomalous Behaviour Analysis

Brenner [14] proposed a definition of anomalous codeposition in which the least noble Fe^2+^ preferentially deposits to the most noble Ni^2+^, and the deposition rate of the medium noble Co^2+^ lies between them. The present experimental results obey the feature of anomalous codeposition. The degree of anomaly for ternary alloy codeposition can be described using the composition ratio value (CRV), as shown in Equation (10) [15,17,49]:(10)$$\mathrm{CRV}=\frac{\mathrm{Fe}\left(\mathrm{Co}\;\mathrm{or}\;\mathrm{Ni}\right)\;\mathrm{mass}\;\mathrm{fraction}\;\mathrm{in}\;\mathrm{deposits}}{{\mathrm{Fe}}^{2+}\left({\mathrm{Co}}^{2+}{\;\mathrm{or}\;\mathrm{Ni}}^{2+}\right)\;\mathrm{bulk}\;\mathrm{concentration}\;\mathrm{in}\;\mathrm{electrolyte}}$$

The degree of anomalous codeposition of Fe, Co, and Ni under different applied current densities is shown in Figure 9. It can be clearly seen that the iron content in the deposits was more than that in the electrolyte, while the cobalt and nickel content in the deposits was less than that in the electrolyte. Meanwhile, the CRV of Co was still higher than that of Ni under the different applied current densities. The widely stated mechanism of anomalous codeposition is that the formation of absorbed mixed-metal intermediates of FeNi(III)_ad_, FeCo(III)_ad_, and CoNi(III)_ad_ [19] or metallic hydroxides M(OH)^−^ [49], depending on the deposition conditions, inhibits the discharging and deposition rate of more noble species. In addition, with increasing applied current density, the CRV of Fe increased slightly, while the CRV of Co decreased sharply, and the CRV of Ni remained steady at a relatively low current density and then increased gradually.

Concentrations of ionic species in the vicinity of the cathode surface are important for electrodeposition [61]. Figure 10 describes the metal ions and hydrogen atom distribution near the cathode surface. It can be observed that the concentrations of Co^2+^ and Ni^2+^ decreased with increasing applied current density. It was also found that the Co^2+^ and Ni^2+^ concentrations near the cathode surface were much lower than those in the electrolyte, while the Fe^2+^ concentration profile showed the reverse trend. At higher current densities, the hydrogen evolution reaction was more severe. Thus, the cathode surface was covered by hydrogen atoms, which in turn blocked the adsorption of metal ions [65]. Furthermore, the ion near the cathode surface was depleted at a rapid rate [15] as the applied current density increased; therefore, less cobalt and nickel were deposited. In contrast, industrial pure iron was employed as a soluble anode, and the Fe^2+^ in the electrolyte was able to be replenished. Moreover, the preferential adsorption of iron intermediates prevented the deposition of nickel and cobalt [65,66,67]. Therefore, the iron concentration near the cathode surface was higher than that in the electrolyte and increased with increasing applied current density.

## 5. Conclusions

A gradient-structured ternary Fe-Co-Ni alloy coating was electrochemically deposited on a steel substrate, and the electrodeposition process was experimentally and numerically investigated. Some conclusions have been drawn as follows: The electrodeposition process was simulated based on the tertiary current distribution. Although there was a deviation between the simulated and experimental results in terms of the coating thickness and elemental contents, the numerical model was still able to predict the variation trend for thickness and element composition and to facilitate the design of the electrodeposition parameters.Gradient-structured coatings were electrochemically fabricated using alternating current and direct current in sequence. Due to hydrogen evolution, microcracks perpendicular to the substrate occurred at the cross-section of the coating, and the network crack was distributed on the coating surface. The XRD patterns show that the deposited coatings consisted of the solid-solution phases α-Fe and γ(Fe, Ni) and the metallic compound Co_3_Fe_7_; the current density in the present studied range had a small influence on the phase composition. The grain sizes on the coating surface varied from 15 nm to 20 nm, and the current density showed no noticeable effect on the grain size. The microhardness of the deposited coatings ranged from 625 HV to 655 HV. Meanwhile, the average microhardness increased slightly as the current density increased from 5 A/dm^2^ to 10 A/dm^2^ and then decreased when the current density further increased.The CRV was calculated to analyse the degree of anomaly. With increasing applied current density, the CRV of Fe increased slightly, while the CRV of Co decreased sharply, and the CRV of Ni remained steady at a relatively low current density and then increased gradually. Finally, the metal ions and hydrogen atom concentrations in the vicinity of the cathodic surface were numerically calculated. With increasing applied current density, the concentrations of Co^2+^ and Ni^2+^ decreased, while the Fe^2+^ and hydrogen atom concentration profiles showed the reverse trend.

## Figures and Tables

**Figure 1 materials-15-06141-f001:**
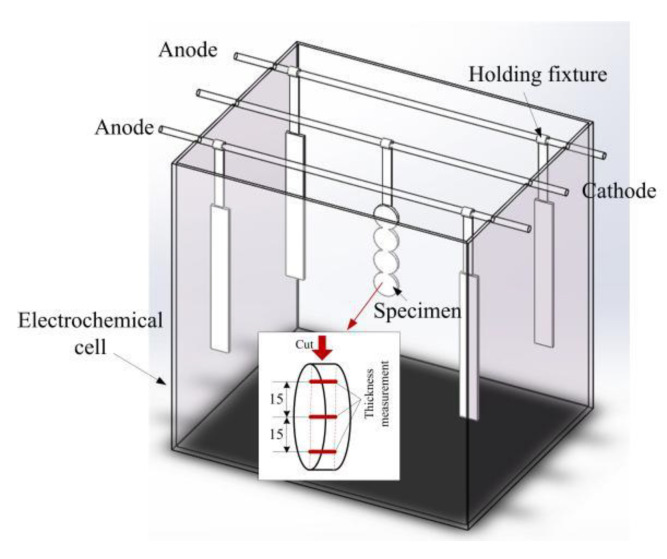
Electrodeposition setup.

**Figure 2 materials-15-06141-f002:**
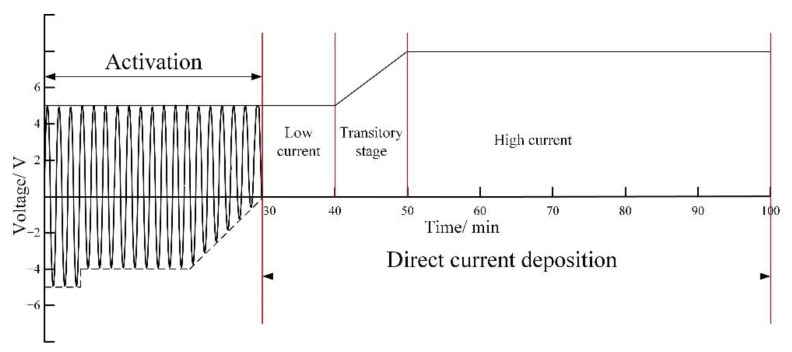
Diagram of electrical control parameter.

**Figure 3 materials-15-06141-f003:**
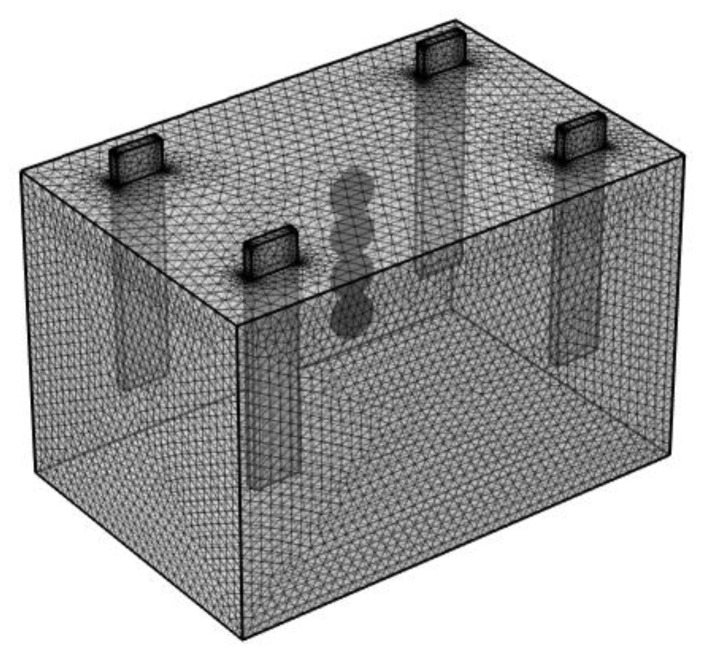
Geometric model and mesh of electrodeposition.

**Figure 4 materials-15-06141-f004:**
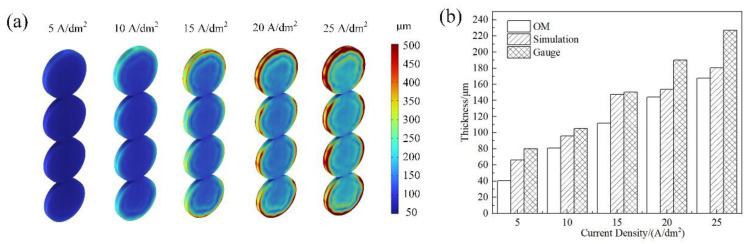
(**a**) Simulated coating thickness at different current densities and (**b**) comparison between simulation and testing results.

**Figure 5 materials-15-06141-f005:**
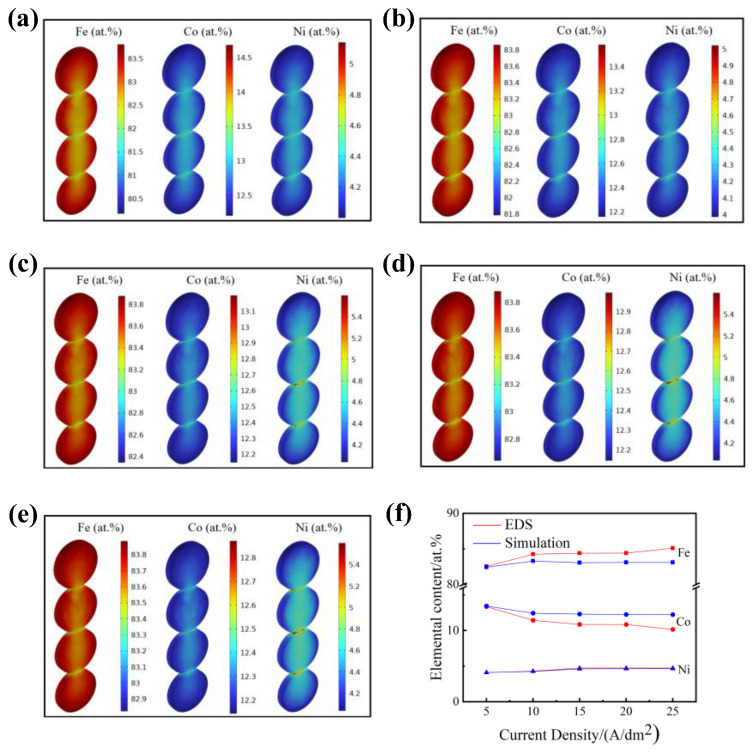
Simulated elemental content in electrodeposited coatings at current densities of (**a**) 5 A/dm^2^, (**b**) 10 A/dm^2^, (**c**) 15 A/dm^2^, (**d**) 20 A/dm^2^, and (**e**) 25 A/dm^2^; (**f**) comparison between simulation and experimental composition.

**Figure 6 materials-15-06141-f006:**
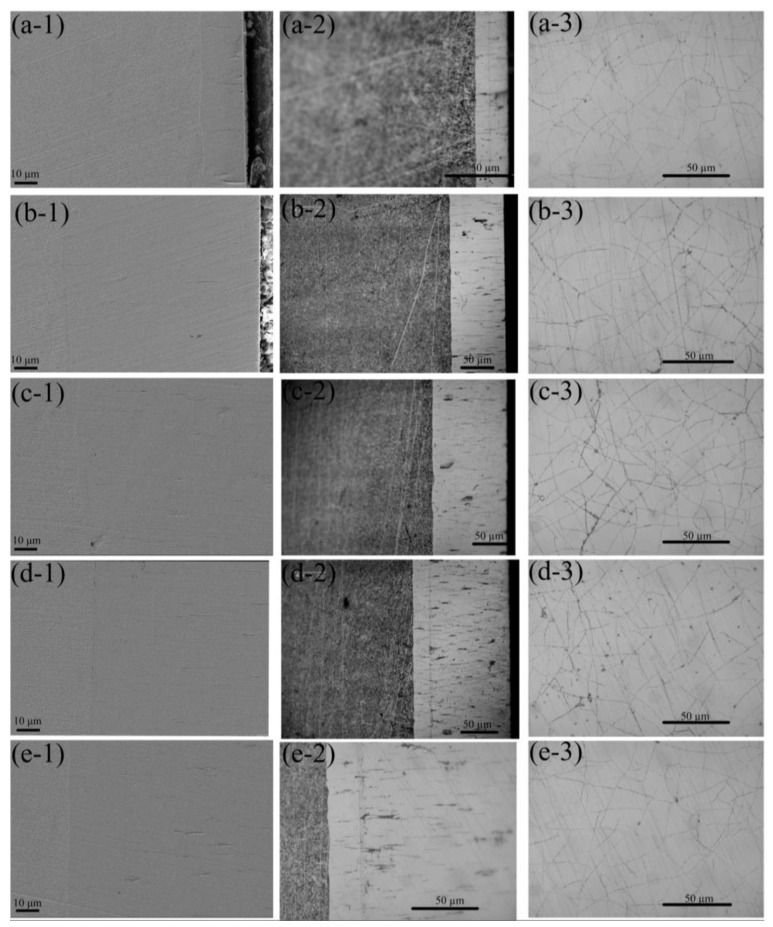
Morphology of cross-section and cracked coating surface deposited at (**a-1**–**a-3**) 5 A/dm^2^, (**b-1**–**b-3**) 10 A/dm^2^, (**c-1**–**c-3**) 15 A/dm^2^, (**d-1**–**d-3**) 20 A/dm^2^, and (**e-1**–**e-3**) 25 A/dm^2^.

**Figure 7 materials-15-06141-f007:**
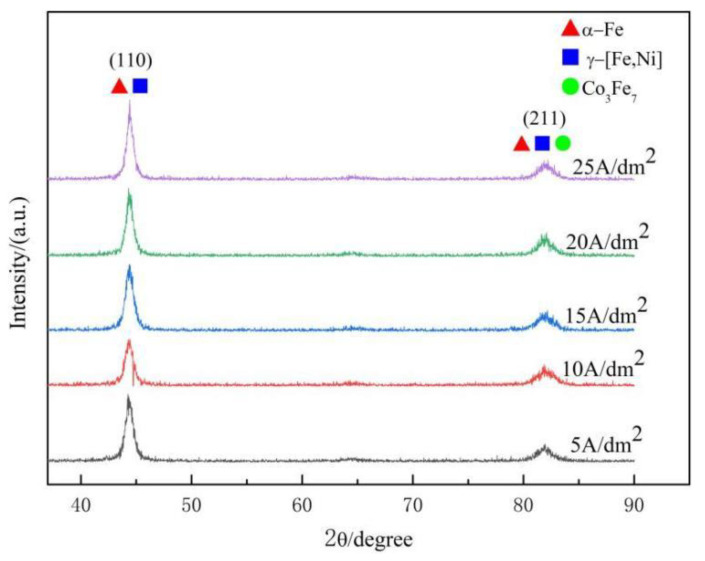
XRD spectra for Fe-Co-Ni electrodeposited coatings at different current densities.

**Figure 8 materials-15-06141-f008:**
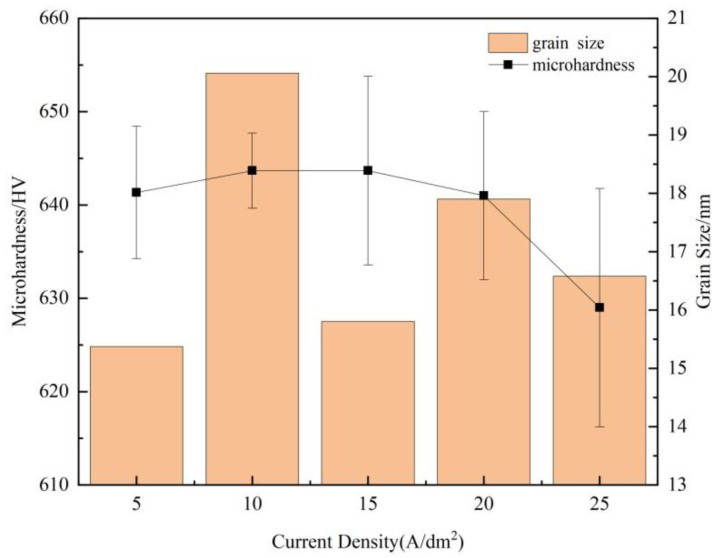
Grain size and microhardness of ternary alloy coatings at different current densities.

**Figure 9 materials-15-06141-f009:**
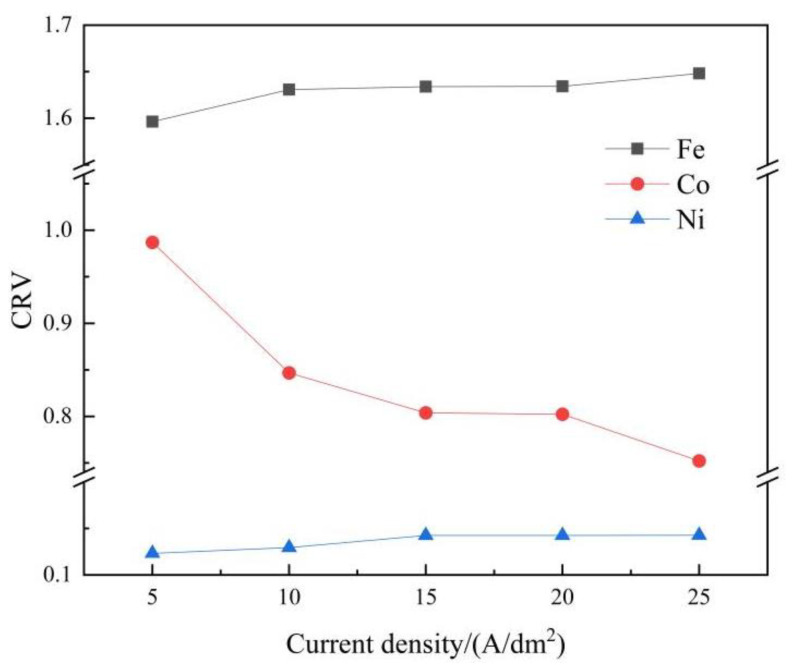
Evolution of the CRV of Fe, Ni, and Co according to applied current density.

**Figure 10 materials-15-06141-f010:**
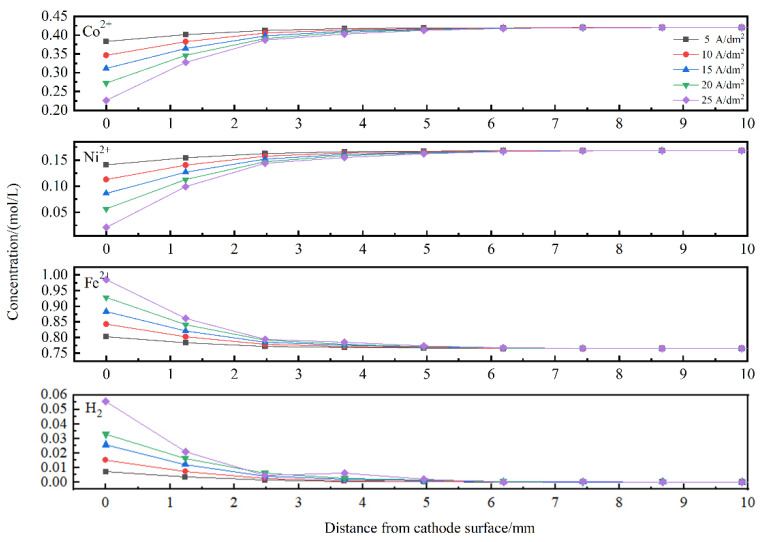
Concentration profiles near cathode surface (*t* = 60 min).

**Table 1 materials-15-06141-t001:** Tertiary current distribution-related equations.

Research Objects	Governing Equations	Parameter Explanations and Notes
Electrodes	$j_{s}=-\sigma_{s}\nabla \phi_{s}$ $\nabla \cdot j_{s}=0$	$j_{s}$ $\;\mathrm{is}\;\mathrm{the}\;\mathrm{current}\;\mathrm{density}\;\mathrm{on}\;\mathrm{the}\;\mathrm{electrode}\;\mathrm{surface},\;\sigma_{s}$ $\;\mathrm{is}\;\mathrm{the}\;\mathrm{electrode}\;\mathrm{conductivity},\;\mathrm{and}\;\phi_{s}$ is the electrode potential.
Electrolyte	$j_{l}=F\underset{m}{\sum }z_{m}N_{m}$ $\nabla \cdot j_{l}=Q_{l}$	$j_{l}$ $\mathrm{is}\;\mathrm{the}\;\mathrm{current}\;\mathrm{density}\;\mathrm{of}\;\mathrm{electrolyte},\;z_{m}$is the charge of species, and F$\;\mathrm{is}\;\mathrm{the}\;\mathrm{Faraday}\;\mathrm{constant};\;N_{m}$$\mathrm{is}\;\mathrm{the}\;\mathrm{ion}\;\mathrm{species}\;\mathrm{flux}\;\mathrm{in}\;\mathrm{electrolyte},\;\mathrm{and}\;Q_{l}$ is electric flux.
Electrode interface	$\eta_{m}=\phi_{s}-\phi_{l}-E_{eq,m}$	$\eta_{m}$ $\;\mathrm{is}\;\mathrm{the}\;\mathrm{surface}\;\mathrm{overpotential},\;\phi_{l}$ $\;\mathrm{is}\;\mathrm{the}\;\mathrm{electrolyte}\;\mathrm{potential},\;\mathrm{and}\;E_{eq,m}$ is the equilibrium potential.
Local current density	$j_{loc,m}=j_{m}\left(\frac{c_{m,red}}{c_{m}}\exp\left(\frac{\alpha_{a}F\eta_{m}}{RT}\right)-\frac{c_{m,Ox}}{c_{m}}\exp\left(\frac{-\alpha_{c}F\eta_{m}}{RT}\right)\right)$	$j_{loc,m}$$\;\mathrm{is}\;\mathrm{the}\;\mathrm{local}\;\mathrm{current}\;\mathrm{density}\;\mathrm{of}\;\mathrm{metal}\;\mathrm{species},\;j_{m}$$\;\mathrm{is}\;\mathrm{the}\;\mathrm{exchange}\;\mathrm{current}\;\mathrm{density}\;\mathrm{of}\;\mathrm{metal}\;\mathrm{species},\;\mathrm{and}\;c_{m,red}$$\;\mathrm{and}\;c_{m,Ox}$ are the species $\mathrm{concentration}\;\mathrm{in}\;\mathrm{deposits}\;\mathrm{and}\;\mathrm{on}\;\mathrm{the}\;\mathrm{electrode}\;\mathrm{surface},\;\mathrm{respectively};\;c_{m}$ $\mathrm{is}\;\mathrm{the}\;\mathrm{species}\;\mathrm{concentration}\;\mathrm{in}\;\mathrm{the}\;\mathrm{electrolyte},\;\mathrm{and}\;\alpha_{a}$$\;\mathrm{and}\;\alpha_{c}$ are the transfer coefficients of anode and cathode, respectively; F$\;\mathrm{is}\;\mathrm{the}\;\mathrm{Faraday}\;\mathrm{constant},\;\eta_{m}$ is overpotential on the electrodes, R is universal gas constant, and T is temperature.

**Table 2 materials-15-06141-t002:** Physical parameters and initial conditions.

Symbol	Terminology	Value	Unit
*Eeq_Fe*	Iron equilibrium potential	−0.444	V
*Eeq_Co*	Cobalt equilibrium potential	−0.290	V
*Eeq_Ni*	Nickel equilibrium potential	−0.276	V
*j* _0_	Current density	5~25	A/dm^2^
*j_Fe_*	Exchange current density of iron [39]	2.09	A/m^2^
*j_Co_*	Exchange current density of cobalt [40]	0.23	A/m^2^
*j_Ni_*	Exchange current density of nickel [29]	0.1	A/m^2^
*j_H_*	Exchange current density of hydrogen [26]	2 × 10^−5^	A/m^2^
*ρ* * _Fe_ *	Density of iron	7900	kg/m^3^
*ρ* * _Co_ *	Density of cobalt	8900	kg/m^3^
*ρ* * _Ni_ *	Density of nickel	8910	kg/m^3^
*M_Fe_*	Molar mass iron	56	g/mol
*M_Co_*	Molar mass of cobalt	58.93	g/mol
*M_Ni_*	Molar mass of nickel	58.693	g/mol
*c_Fe_*	Fe^2+^ concentration	0.765	mol/L
*c_Co_*	Co^2+^ concentration	0.42	mol/L
*c_Ni_*	Ni^2+^ concentration	0.168	mol/L
*z_Fe_*	Charge of iron	2	-
*z_Co_*	Charge of cobalt	2	-
*z_Ni_*	Charge of nickel	2	-
*D_Fe_*	Diffusion coefficient of iron	1.59 × 10^−9^	m^2^/s
*D_Co_*	Diffusion coefficient of cobalt	1.31 × 10^−9^	m^2^/s
*D_Ni_*	Diffusion coefficient of nickel	1.27 × 10^−9^	m^2^/s
*A*	Cathode surface area	1	dm^2^
*σ*	Electrolyte conductivity [29]	10	S/m
*T* _0_	Temperature	348.15	K
*A_c_*	Tafel slope for hydrogen evolution [41]	−188	mV
*α_a_*	Anode transfer coefficient [41]	0.5	-
*α_c_*	Cathode transfer coefficient [41]	0.5	-

**Table 3 materials-15-06141-t003:** Boundary conditions.

Boundary (Surface)	*V*	*A*	*c*
Electrochemical cell	−n·j=0	−n·i=0	-
Bath	$-n\cdot j=\phi_{l}$	$-n\cdot i=j_{0}A$	-
Anode	$-n\cdot j=\phi_{s}$	$n\cdot i=j_{0}A$	$-\frac{dc_{s,m}}{dt}=-\nabla \cdot N_{m}+R_{m}$
Cathode	$-n\cdot j=-\phi_{s}$	$n\cdot i=-j_{0}A$	$\frac{dc_{s,m}}{dt}=-\nabla \cdot N_{m}+R_{m}$

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
