# Peer review of "Numerical and Experimental Investigation of the Effect of Current Density on the Anomalous Codeposition of Ternary Fe-Co-Ni Alloy Coatings"

_materials, 2022, doi:10.3390/ma15176141_

Round 1

Reviewer 1 Report

The present manuscript: “Numerical and experimental investigation of the effect of current density on the anomalous deposition of ternary Fe-Co- 3 Ni alloy coatings”; studies the deposit of a ternary alloy through numerical simulation and experimental analysis, showing how this can be an excellent tool to save experimental tests and as these results show a good correlation with the experimental results.

The manuscript presents a good structure and makes a good discussion of the results, showing only minor deficiencies that must be resolved, which are indicated below:

1. Line 77 to 96: The introductory part of the simulation needs to be structured to understand the importance of the contributions that other works have made.  As written, it seems like a list of those who have done simulation without discussing why they do it or how those works contribute to this study.

2. Change the letter “L” to “l” in Figure 1 at the word “Electrochemical cell.”

3. Clarify in section 2.1 the electrodeposition methodology since it is confusing when showing a diagram as a function of the potential when the treatments are galvanostatic as implied in the results, redo figure 2. Placing how you made those different states identifies the current density values used.

4. Pointing out what you want to show the reader in figure 6 in all sections 1 (a-1, b-1,..., e-1) is unclear.

Author Response

1. The discussion of research status of simulation is added in Line 100 to 106, page 3.

Although many scholars have developed numerical models of electrodeposition, simplified one step electrochemical reaction is used in the above simulation. However, for the electrodepostion of ternary Fe-Co-Ni alloys, the reaction process is an irreversible two-step manner. In addition, the side reactions of hydrogen evolution and water dissociation have important influences on the properties of deposits. To the best of our knowledge, there is no specific simulation study of Fe-Co-Ni ternary alloy deposition.

2. The misspelling in Fig. 1 is corrected.

3. Fig. 2 is redrawn, and the current density value is given in Line 140 to 141, page In this study, we adjusted only the high current stage to investigate the influence of the current density. The applied current density in high current stage is 5 A/dm2, 10 A/dm2, 15 A/dm2, 20 A/dm2, and 25 A/dm2.

4. Fig. 6 shows the influence of current density on the morphology of cross section and coating surface of ternary Fe-Co-Ni alloy coatings. This sentence is added in Line 270 to 271, page 10.

Reviewer 2 Report

Title: Numerical and experimental investigation of the effect of current density on the anomalous codeposition of ternary Fe-Co-Ni alloy coatings

Authors: Shuai Zhang, Jing Yu, Zhengda Liu, Yanjun Yin, Chenfeng Qiao

In this paper, Zhang and coworkers, have studied the properties of the ternary Fe-Co-Ni alloy coatings electrodeposited on steel substrates at various current densities from chloride baths. The as obtained coatings have been characterized by using the SEM, EDX and XRD. The microhardness of the coating surface was measured. The results obtained experimentally have been used to verify the numerical model.

  • The structure of the article fulfills the structure of a research article.

o   For keywords are included by the author.

  • The Introduction section provide sufficient background information for readers in the immediate field to understand the problem that this study addresses.

I suggest to Reconsider after Minor Revisions for the following reasons:

1. the 0.4 to 0.6 pH value of the electrochemical bath is too low. Please check the pH value or add a reference.

2. page 11, line 303: replace “monocrystalline” with “nanocrystallites”;

3. the English language should be carefully corrected.

Author Response

1. For the electrodepostion of Fe-based alloy at low temperature (30-50℃), the electrolyte pH is very important and must be adjusted carefully. In the previous study from our research group, binary Fe-based alloy coating was successfully fabricated under the bath pH of 0.8 to 1[11,12]. For the electrodeposition of ternary FeCoNi alloy from simple chloride bath, the electrolyte pH must maintain at very low value (0.5 to 1.5) based on Electroplating Manual (China Machine press, April 2010. in Chinese). We conducted extensive experiments, and found that the deposited coating of high quality was obtained at the pH value of 0.4 to 0.6.

[11] XINTAN H. Crankshaft restoration by iron plating. Metal finishing, 1997, 6(95): 98-100.

[12] R.X.Huang, Z.Ma, W.Z.Dong, Y. Shen, F.M.Du, J. Xu, M. Jin. On the Adhesive Strength Quantification and Tribological Performance of the Multilayered Fe–Ni Coating Fabricated by Electroplating. Strength of Materials, 2019, 51(2): 280-290.

2. The misspelling was corrected.

3. We carefully double-checked the spelling, grammar, sentence structure and terminology in the paper.

Our paper has been performed a detailed proofreading by a professional English editing company - AJE (American Journal Experts), and the certification is provided as follows:

Reviewer 3 Report

The paper focuses on the microstructural evolution of electroplated Fe-Co-Ni coatings grown on the surface of a 42CrMo steel using a plating bath containing chlorides of Fe, Co and Ni. A simulation of the electrodeposition process was performed using FEM. The results show that a gradient structure develops within the coating, which is interesting. My comments are as follows:
1) Page 2, line 50: What is the “improved Ollar method”? Do the authors mean “modified Ollard”?
2) Page 8, line 234: “the simulated elemental content seems possibly much even than the experiments”. Please rephrase.
3) What is the difference between Figs. 6a1-e1 and Figs. 6a2-e2? The caption for Fig. 6 needs to be rewritten.
4) In page 9, line 267, the microcracking is explained by the claim that “the grain growth direction is weaker than that along the growth direction”. The reference given for this claim, Ref. 52, makes no mention of grain growth.
5) In page 11, line 290, it is claimed that alpha-Fe is the stable phase for Fe content over 80%. In what alloy?
6) Page 11, line 293: “the solid solution phase α-Fe is supposed to incorporation of cobalt and nickel into iron”.
7) Page 11, line 296: “a preferential orientation generated by electric crystallization”. The cited reference, Ref. 58, does not appear to make such a claim.
8) Page 11, line 318: “microsturectures”.
9) Page 12, line 321: “lest noble”. And how can Fe2+ deposit on an ion?
10) Fig. 9: The axis label “Degree of anomaly (dA)” is not defined in the text.

Author Response

1)Here we mean a modified Ollard method. And we change the expression as reviewer’s suggestion in Line 52.

2)This sentence is rewritten as follows:

Due to the exclusion of convention in the electrolyte and the biphasic effect in the numerical calculation, the elemental content distribution in the deposited coatings is relatively homogeneous, and the elemental contents vary within a reasonable range.
3) Both Fig. (a-1)-(e-1) and Fig. (a-2)-(e-2) show the cross sections of deposited coatings. But Fig. (a-1)-(e-1) images were captured using SEM, while Fig. (a-2)-(e-2) were observed using OM.

The caption for Fig. 6 is rewritten as follows:

Morphology of cross-section and cracked coating surface deposited at (a) 5 A/dm2 , (b) 10 A/dm2 , (c) 15 A/dm2, (d) 20 A/dm2 and (e) 25 A/dm2.
4) We misplaced the reference number. In Ref.52, the growth direction of columnar grains near the interface is investigated.

[52] Bastos, S. Zaefferer, D. Raabe, C. Schuh. Characterization of the microstructure and texture of nanostructured electrodeposited NiCo using electron backscatter diffraction (EBSD). Acta materialia, 2006, 54(9):2451-2462.

5) In Ref. 50, authors mentioned that the bcc α-Fe phase is stable for Fe concentration in excess of 80 at % at temperatures below 500℃. In our present study, the iron content in atom fraction in the deposits is close or beyond 80%.

[50] ZHANG Y, IVEY D G. Characterization of Co-Fe and Co-Fe-Ni soft magnetic films electrodeposited from citrate-stabilized sulfate baths. Materials Science and Engineering: B, 2007, 140(1-2):15-22.

6)  This sentence is rewritten as follows:

α-Fe is supposed to be the substitutional solid solution, which incorporates cobalt and nickel into iron.
7)We misplaced the reference number. In Ref.58, the relationship between the current density and the preferred crystalline orientation is investigated.

[58] THIEMIG D, BUND A, TALBOT J B. Influence of hydrodynamics and pulse plating parameters on the electrocodeposition of nickel–alumina nanocomposite films. Electrochimica Acta, 2009, 54(9): 2491-2498.

We carefully double-checked the reference numbers in the paper.

8) This misspelling is changed.
9) The misspelling is changed.

According to Ref. 19, during NiCoFe ternary alloy deposition, the mixed metal intermediates FeNi(III)ad, FeCo(III)ad, and NiCo(III)ad and single-metal intermediates are formed and adsorbed at the cathode surface. Metal ions are reduced from above intermediates and the preference for high surface coverage by Fe intermediates caused inhibition of the Ni deposition. 

[19] Y Zhuang, E.J. Podlaha. NiCoFe ternary alloy deposition: III. A mathematical model. Journal of the Electrochemical Society, 2003, 150(4): 225-233.

10) The degree of anomaly for ternary alloy codeposition can be described using the composition ratio value (CRV).

The axis label in Fig. 9 is changed.
